# Associations between Hypertension, Treatment, and Cognitive Function in the Irish Longitudinal Study on Ageing

**DOI:** 10.3390/jcm9113735

**Published:** 2020-11-20

**Authors:** Sook Ling Leong, Ian H. Robertson, Brian Lawlor, Sven Vanneste

**Affiliations:** 1Global Brain Health Institute & Trinity Institute of Neuroscience, Trinity College Dublin, DO2 PN40 Dublin, Ireland; sookling.leong@tcd.ie (S.L.L.); iroberts@tcd.ie (I.H.R.); blawlor@stjames.ie (B.L.); 2School of Psychology, Trinity College Dublin, DO2 PN40 Dublin, Ireland

**Keywords:** hypertension, cognition, antihypertensives

## Abstract

Epidemiological studies have produced conflicting results regarding the associations between the use of different hypertensive drugs and cognition. Data from the Irish Longitudinal Study on Ageing (TILDA), a nationwide prospective longitudinal study of adults aged 50 or more years, was used to explore the associations between hypertensive status, categories of antihypertensive and cognitive function controlling for age, education, and other demographic and lifestyle factors. The study sample included 8173 participants. ANCOVAs and multivariate regressions were used to assess the cross-sectional and longitudinal associations between cognitive function and hypertension status and the different categories of hypertensive medication. Hypertension was not associated with decline in global cognitive and executive functions and were fully explained by age and education. Different hypertensive medications were not associated with cognitive function. Consistent with previous studies, changes in cognition can largely be explained by age and education. The use of antihypertensive medications is neither harmful nor protective for cognition.

## 1. Introduction

Hypertension and its potential effects on cognitive function and decline, notably in older adults is a major concern worldwide [1]. It has been reported that, globally, 65–75% of adults over 65 years are hypertensive [2]. With an increase in life expectancy, the global prevalence of dementia is expected to rise to above 80 million by 2040 [1]. 

Mounting evidence suggest that hypertension is an important risk factor for the development of mild cognitive impairment (MCI) and Alzheimer’s disease (AD) [3]. Currently, although the pathophysiological pathways of this association are contentious, most research converges on the contribution of vascular dementia, wherein the combination of aging and uncontrolled high blood pressure impairs neurovascular regulations involved in various cognitive functions [4]. 

To date, epidemiological studies have produced conflicting results regarding hypertension and the development of cognitive decline. For example, a prospective 23-year study of 13,476 individuals with a mean age of 56 years at baseline reported a significant modest association between baseline continuous systolic blood pressure (SBP) and rate of cognitive decline [5]. In a community-based cohort study of 918 individuals aged 65 and above, hypertension, defined categorically as above 140 mm Hg (SBP) and 90 mm Hg (diastolic, DBP), was related to higher risk of developing MCI five years later. Hypertension was also associated with worse executive ability scores, but not with memory or language scores [6]. Conversely, the 13-year longitudinal French SU.VI.MAX study of 2788 participants with a mean baseline age of 65 showed that elevated baseline BP (SBP/DBP ≥130/85 mmHg or antihypertensive medication use) was not associated with cognitive decline at follow-up [7]. 

The effects of different antihypertensive drugs on cognition remain inconclusive. Earlier studies have shown that calcium-channel blockers (CB) and angiotensin-converting enzyme inhibitors (ACE) reduced the risk of developing dementia by 50% and 34%, respectively [8]. On the contrary, the use of beta-blockers (BB) in some studies has been associated with adverse effects on cognitive function [9]. However, a recent systematic review of 358 studies reported that the use of ACEs, BBs, and CBs do not have positive or negative repercussions on cognitive function in older adults [10]. Furthermore, given the class-specific effects of antihypertensive drugs, combination therapies in contrast to monotherapy may provide greater cognitive protection [11]. 

One contributing factor to interindividual variations in cognition during the aging process is cognitive reserve. The term cognitive reserve is routinely used to refer to the causal association between experiential resources (e.g., education, knowledge, healthy diet, physical activity) and cognitive function; whereby higher levels of experiential resources have a protective effect on late-life cognitive decline [12]. Measuring cognitive reserve is challenging and there is, at present, a wide range of methods being employed [13]. Of the different proxies, level of education is one of the most frequently used indicators of cognitive reserve [13]. Studies have provided evidence that more education contributes substantially to building up cognitive reserve [14]. This in turn reinforces an individual’s capacity and resilience to cope with the effects of the aging process on cognition through constructive recruitment of neural networks [14]. 

The compensation effect of cognitive reserve has been reported to offset the harmful consequences of hypertension on cognition [15]. In a cohort of 701 older adults, hypertension (≥ 140 mmHg SBP) was associated with lower performance in cognitive assessments (i.e., immediate and delayed cued recall as well as working memory). Moreover, it was reported that longer education and higher cognitive level of job played important roles in determining the performance during cognitive tasks [15]. These results suggest that early life cognitive reserve may be an important contributor in reducing the detrimental effects of hypertension on cognition [15].

It is noteworthy that age and education are by far the most prominent factors associated with memory decline and impairment during the aging process [16,17,18,19]. Furthermore, reduced memory performance has been associated with diseases such as diabetes and cancer. In addition, studies have shown that lifestyle factors, for example, smoking, alcohol consumption, and physical activity, play important roles. In a recent study of 23,641 European adults aged ≥60 years, severe limitations in physical activities and past alcohol problems were negatively associated with memory performance [19]. These results were in line with previous research reporting that higher levels of alcohol consumption and increased frequency of smoking are related with loss in cortical grey matter, poorer cognition, and memory impairments [20,21,22]. In contrary, healthier lifestyle habits such as increased physical activity is related to improved memory performance in young and older adults [23].

In the current study, we utilized data from the Irish Longitudinal Study on Ageing (TILDA) to explore the associations between (a) hypertensive status and cognitive function, (b) categories of antihypertensive medication and cognitive function, and (c) the interactions between hypertensive status and categories of hypertensive medication with cognitive reserve and their effects on cognitive function. Taking into account that previous literature has consistently shown that impacts of aging and lower levels of education on cognitive function, in order to ascertain the effects of different hypertensive status and medications on cognition [16,17,18,19], all initial models were controlled for age and education. Further, multivariate models were carried out with different lifestyle factors as covariates.

## 2. Methods

### 2.1. Population

Data were from the first (2009–2010) and third waves (2014–2015) of the Irish Longitudinal Study on Ageing (TILDA). TILDA is a nationwide prospective longitudinal study of adults aged 50 or more years in the Republic of Ireland. The final sample of the present analyses included 8173 participants from Wave 1 (W1) and 6248 participants from Wave 3 (W3) who completed home-based computer-assisted personal interviews and physical health assessments at dedicated health centres. This study was ethically approved by the Trinity College Research Ethics Committee, and all related activities adhered to the Declaration of Helsinki. All participants provided signed informed consent preceding participation. Features of the TILDA cohort and its design has been detailed elsewhere [24]. 

### 2.2. Hypertension and Cognitive Function

Mean seated blood pressure (BP) from two measurements were obtained using a digital automated oscillometric BP monitor (Omron M10-IT, Omron Inc., Kyoto, Japan). Occurrence of hypertension was defined as systolic BP ≥ 140 mmHg or diastolic BP ≥ 90 mmHg and/or currently taking antihypertensive medications. Antihypertensive medication was categorized in accordance with the World Health Organization (WHO) Anatomical Therapeutic Chemical (ATC) classification system. 

Global cognitive function was assessed using the Mini-Mental State examination (MMSE) [25] and the Montreal Cognitive Assessment (MoCA) [26]. MMSE has a maximum score of 30 with a of cut-off of below 24 indicating cognitive problems. Scores on the MoCA ranges from 0 to 30 with a score of 26 and higher to be considered normal. The Color Trails Test (CTT) [27] measured sustained attention. In part 1 (CTT-1) respondents connected circles number 1–25 in sequence. In part 2 (CTT-2) respondents connected numbers in sequence but alternated between pink and yellow. A shorter length of time to complete this test reflects greater executive functioning. The National Adult Reading Test (NART) [28] was administered at W3 as a proxy of cognitive reserve. The degree of literacy is measured by the number of correct responses to reading 50 irregular words, arranged from simple to difficult. The NART is a widely used proxy for cognitive reserve and its validity has been previously established [29]. Higher NART IQ has been associated with greater cognitive research capacity [12,30].

### 2.3. Socio-Demographic and Health Characteristics

Socio-demographic and health characteristics used as covariates in this study included age (continuous variable), sex (male and female), highest educational attainment (primary, secondary, and tertiary), current employment status (currently employed, retired and other), current smoking status (non-smoker, past smoker, and current smoker), being on antidepressants (not taking antidepressants and taking antidepressants), moderate physical activity (≥ 5 times/week and < 5 times/week) [31], alcohol use disorder (yes—having alcohol problem and no—not having alcohol problem) [32] diabetes (yes—have diabetes, no—do not have diabetes) and systolic and diastolic blood pressures (continuous variables). The International Physical Activity Questionnaire [31] was used to classify <5 times of moderate activity/week and ≥5 times of moderate activity/week. Problematic alcohol intake was measured as a score of ≥2 on the Cut, Annoyed, Guilty, Eye-opener (CAGE) questionnaire [32]. 

### 2.4. Statistical Analysis

Hypertension status was categorized into three groups: Non-hypertensive (H1), Hypertensive without medication (H2) and Hypertensive with medication (H3). In addition, those who were on monotherapy were classified into: Beta-blockers (BB), Calcium-channel blockers (CB), and Angiotensin-converting enzyme inhibitors (ACE). Participants in the H3 group were further divided into combination therapy (i.e., taking more than one medication) and those who were on monotherapy (taking only one medication).

All models were initially assessed for equal variances using the Levene’s test [33] and the Breusch–Pagan [34] test for homoskedasticity. Where assumptions were violated, regression models instead of analysis of co-variance (ANCOVAs) were developed correcting with the robust Huber–White sandwich estimator of variance [35]. Analysis of variance (ANOVA) was used to examine the cross-sectional differences in socio-demographic and health characteristics at W1 and W3 by hypertension status.

To examine 1. hypertensive status (H1, H2, and H3) and cognitive function and 2. categories of antihypertensive medication (BB, CB, and ACE) and cognitive function, ANCOVA with age and education levels as covariates were conducted in partial models (Figure 1a) and multivariate regressions were utilized with all socio-demographic and health characteristics as confounders in full models (Figure 1b). Changes in cognitive measures between W1 and W3 were used as dependent variables in longitudinal ANCOVA models, while in multivariate models, W3 dependent outcomes were used controlling for their corresponding W1 scores. In the investigation of 3. medication status (monotherapy and combination therapy) and cognitive function, multivariate models were used in both the partial (controlling for age and education) (Figure 1a) and full models (controlling for all potential confounders) (Figure 1b). In the longitudinal models, scores of W3 dependent outcomes were used controlling for W1 corresponding scores. To examine 4. the interactions between hypertensive status (H1, H2 and H3), categories of hypertensive medication (BB, CB, and ACE) and medication status (monotherapy and combination therapy) with cognitive reserve (NART) and their effects on cognitive function (Figure 1c), multivariate models were utilized in the partial and full models). Given that NART was only assessed at W3, this variable was examined cross-sectionally. 

The effect sizes of each variable are reported using Cohen’s d. All analyses were conducted using STATA 14.0 (Stata Corp, College Station, TX, USA). Where two-side *p* < 0.05, the associations between variables were considered statistically significant.

## 3. Results

### 3.1. Socio-Demographic and Health Characteristics

Cross-sectionally, at W1 (Table 1) and W3 (Table 2), H1 participants were younger, attained a higher level of education, were currently employed, were never or past smokers, and were not diabetic compared to H2 and H3.

### 3.2. Hypertension Status (H1, H2 and H3)

Cross-sectionally, at W1 and W3 (Table 3), in the partial models, there were no evidence of a significant effect of hypertension status (H1, H2 and H3) on MoCA (W1, *p* = 0.3052; W3, *p* = 0.5957), CTT-1 (W1, *p* = 0.3579; W3, *p* = 0.4752), and CTT-2 (W1, *p* = 0.4090; W3, *p* = 0.1383). Similar results were observed in the longitudinal model (Table 3). Hypertension status had a significant effect on MMSE in W3 (*p* = 0.0009) but not at W1 (*p* = 0.8900) and when examined longitudinally (*p* = 0.5806) (Table 3). Bonferroni post hoc analyses for cross-sectional and longitudinal models showed that younger and more educated individuals scored significantly higher (*p* < 0.0001) across these different cognitive measures. 

Multivariate regression results for the full models indicated that aside from age and education, lifestyle and other demographic factors significantly contributed to the impact of hypertension status on cognition, MMSE (Appendix A), MoCA (Appendix A), CTT-1 (Appendix A), and CTT-2 (Appendix A). Cross-sectionally and longitudinally, Cohen’s d showed that, while age and education explained most of the model’s variances (5% to 15%), lifestyle and demographic factors played significantly (*p* < 0.05) important roles in determining an individual’s cognitive outcome (Appendix A).

### 3.3. Categories of Medication (BB, CB and ACE)

Similar to results of the effects of hypertensive status on cognitive measures, the different categories of single medication (i.e., BB, CB, ACE) were not significantly related to MMSE (W1, *p* = 0.6247; W3, *p* = 0.4168; Longitudinal, *p* = 0.0727), MoCA (W1, *p* = 0.7211; W3, *p* = 0.8435; Longitudinal, *p* = 0.8502), CTT-1 (W1, *p* = 0.9972; W3, *p* = 0.0740; Longitudinal, *p* = 0.9656), and CTT-2 (W1, *p* = 0.6114; W3, *p* = 0.1506; Longitudinal, *p* = 0.8551) after controlling for age and education (Table 4). Post-hoc Bonferroni showed that younger and more educated individuals scored significantly higher (*p* < 0.0001) across these different cognitive measures. 

In cross-sectional (W1, W3) and longitudinal multivariate regression models, besides age and education, lifestyle and demographic factors were significantly (*p* < 0.05) associated with the different cognitive measures (Appendix A). When these variables were included in the full models, there were no significant associations between MMSE (Appendix A), MoCA (Appendix A), CTT-1 (Appendix A) and CTT-2 (Appendix A) with the different single use medication categories cross-sectionally and longitudinally.

### 3.4. Medication status(Monotherapy and Combination Therapy)

Interestingly, at W1 compared to monotherapy, combination therapy was significantly associated with the different cognitive measures after accounting for age and education in the partial models: MMSE; β = 0.04, t = 1.99, *p* = 0.04, MoCA; β = 0.05, t = 2.52, *p* =0.012, CTT-1; β = −0.04, t = 1.33, *p* = 0.040, CTT-2; β = −0.04, t = 1.99, *p* = 0.042 (Table 5). These significant results were however not observed at W3 and longitudinally (Table 5).

At W1, the effects of medication status on MMSE, MoCA, CTT-1 and CTT-2 were no longer significant when lifestyle variables were included in the full models (Appendix A). Results of multivariate regression models demonstrated that aside from age and education, lifestyle and demographic variables play significantly (*p* < 0.001) important roles in determining the level of cognitive function in individuals when observed cross-sectionally and longitudinally for MMSE (Appendix A), MoCA (Appendix A), CCT-1 (Appendix A), and CTT-2 (Appendix A).

### 3.5. NART

At W3, in the partial (controlling for age and education) and full (controlling for age, education, and other lifestyle and demographic factors) models, the interaction between the different categories of hypertension status (H1, H2, H3) and NART on MSSE, MoCA, CTT-1, and CTT-2 were not significant (Table 6). 

There were no significant interactions between the different medication categories (BB, CB, ACE) and NART on MMSE, MoCA, CTT-1, and CTT-2 at W3 in both the model controlling for age and education as well as the full model controlling for demographic and lifestyle factors (Table 7).

In the partial model, results indicated a significant interaction between medication status (monotherapy and combination therapy) and NART on MMSE (*p* = 0.024) and CTT-2 (*p* = 0.012). However, these interactions were no longer significant in the full model controlling for age, education, and other demographic and lifestyle factors (Table 8).

## 4. Discussion

The main aim of this study was to examine the associations between (a) hypertensive status and cognitive function, (b) categories of antihypertensive medication and cognitive function, and (c) the interactions between hypertensive status and the different categories of antihypertensive medication with cognitive reserve and their effects on cognitive function. All initial models were controlled for age and education while other demographic and lifestyle factors were included in subsequent multivariate analyses. In this cohort, hypertension was not associated with decline in the global cognitive function measures of MMSE and MoCA after controlling for age and education. Similar results were observed in measures of executive function (CTT-1 and CTT-2). Study results also demonstrated no differences in cognitive function between participants who were on BB, CB, or ACE in the initial models. In addition, being on combination therapy or monotherapy had no impact on cognition after controlling for age and education. Further, MMSE, MoCA, CTT-1, and CTT-2 scores were not influenced by interactions between hypertension status and the categories of hypertensive treatments with NART in multivariate models.

At present, epidemiological studies evaluating the relationship between BP and cognition in older adults have produced inconsistent results, with studies showing beneficial, harmful, and a lack of effect of hypertension on cognitive function [36]. Heterogeneity in study findings could result from variations in cognitive function measures, blood pressure estimates, cohort differences and study durations [36]. Findings from this study highlights that education and age had modest effect sizes in the multivariate models. For MMSE and MoCA, education seems to play a more important role in explaining model variances cross-sectionally while age, longitudinally. These results, together with previous studies [37] converges towards evidence that, although education may have a rather small effect on cognitive decline over time, those with higher educational attainment will consistently have better cognitive performance as they age as a consequence of the passive model of cognitive reserve [37]. 

In this study, when examined cross-sectionally, the moderating effect of NART on MMSE, MoCA, CTT-1 and CTT-2 could be explained—by age and education. Results from TILDA for MMSE and MoCA stratified by age and education have been published [38,39,40]. Despite these negative findings regarding NART, it is important to note that results are cross-sectional, making it impossible to distinguish the true effects of NART over time. Indeed, it has been previously reported that NART significantly predicts changes in episodic and working memory over a 7-year period after controlling for age, level of education, and socio-economic status [41]. 

Concerning usage of hypertensive medication, there was a lack of evidence of an effect on cognition when comparing combination therapy to those on monotherapy as well as the different types of hypertensive treatments. These results, consistent with previous epidemiological studies, underscore the need for controlled randomized trials investigating the influence of antihypertensives on cognition given the heterogenous nature of population-based studies [36]. A recent meta-analysis of over 50,000 participants aged >65 years from 27 studies, including the TILDA, reported a lack of risk on cognitive decline for BB, ACE and CB [42]. 

Interestingly, in subsequent models, health and lifestyle factors were overall significant in cross-sectional analyses but not longitudinally. Cross-sectional results are consistent with previous studies reporting that healthier lifestyle habits: moderate physical activity, non-smokers, and not having a drinking problem are associated with better cognitive measure outcomes [16,43]. However, in longitudinal models, baseline age and education accounted for variances. The use of categorical instead of continuous assessments for demographic and lifestyle covariates have simplified analyses and interpretation. Yet, this could have reduced statistical power to detect relations between the independent and dependent variables, increased false positive results, underestimate variations in outcome between groups and assumed non-linear associations [44].

Several limitations in this study are notable. The cohort in this study, when categorized into different subgroups had average normal MMSE and MoCA scores at baseline. At Wave 1 of the TILDA study, potential participants who had or were suspected of having dementia were excluded during enrolment [39,45]. This contributes to underestimation of the associations between the different subgroups and cognition, limiting the generalizability of these results. Furthermore, the use of self-reported assessment for hypertension medication meant that adherence was not accounted for. Furthermore, previous use and length of treatment before enrolment could have introduced bias to study results. 

It is important to address the decision to include both the MMSE and MoCA in these analyses. Although MMSE is the most commonly used test for cognitive screening, the MoCA has been shown to have higher sensitivity and specificity in detecting MCI or dementia [46,47,48,49]. The MoCA discriminates well between normal cognition and mild MCI or dementia while the MMSE is likely a better assessment for more severe conditions [46,47,48,49]. It is also crucial to note that the MMSE and MoCA are best utilized in clinical settings to categorically diagnose patients with or without dementia and when applied to a more general population, may be insensitive in capturing change over time [46,47,48,49]. The categorical nature of these measures may introduce the risk of a ceiling effect as a result of the inability of the scale to include true values and variability above the maximum value. In addition, the NART was developed to assessed premorbid intelligence among dementia patients in clinical settings [50], thus limiting its usability in the general population. Therefore, results of this study should be interpreted with caution as they are susceptible to false negatives, given that dependent outcome measures may not truly reflect the range of cognitive decline in a healthy aging population.

The large sample size that included a national sample and the use of objective blood pressure measurements are strengths of the study. The large number of variables collected allowed us to control for a multitude of potential confounders in the multivariate models. Our results highlighted the importance of taking other socio-demographic and health behaviour characteristics into account, given that the magnitude of change in cognition resulting from the process of aging is moderated by lifestyle factors encompassing physical activity, alcohol intake and smoking status [51]. 

## 5. Conclusions

In conclusion, in this national population-based cohort of aging adults, there was a lack of independent association between hypertension and the use of antihypertensive medications on cognition. Results suggest that changes in cognition as measured by MMSE, MoCA, and the Color Trial Tests can largely be explained by education cross-sectionally and age longitudinally. Findings also indicate that the use of combination therapy or the different single antihypertensive medication are neither harmful nor protective for cognitive functioning.

## Figures and Tables

**Figure 1 jcm-09-03735-f001:**
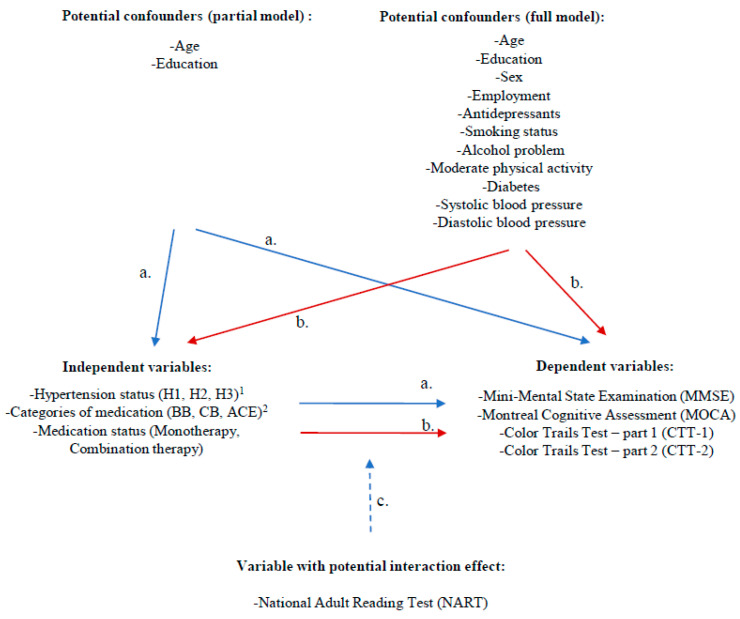
Analyses model investigating the associations between hypertension, categories of medication, medication status, and cognitive outcome measures cross-sectionally and longitudinally controlling for potential confounders and interaction effect. a. Potential confounding variables in the partial models. b. Potential confounding variables in the full model. c. NART was used to examine the potential interaction effect at W3 between the independent and dependent variables. ^1^ H1 = Non-hypertensive, H2 = Hypertensive without medication, H3 = Hypertensive with medication. ^2^ BB = Beta-blockers, CB = Calcium-channel blockers, ACE = Angiotensin-converting enzyme inhibitor.

**Table 1 jcm-09-03735-t001:** Comparison of socio-demographic and health characteristics of participants at Wave 1 using analysis of variance (ANOVA).

	H1 ^1^	H2 ^2^	H3 ^3^	*p*-Value
**Age, years** (Mean, SE)	59.5 (0.16)	62.7 (0.18)	68.1 (0.18)	<0.0001
*n*	*2280*	*2823*	*3070*	
**Blood pressure** (Mean, SE)				<0.0001
Systolic (mm Hg)	121.5 (0.23)	152.6 (0.38)	139.3 (0.43)	
Diastolic (mm Hg)	76.5 (0.16)	92.1 (0.25)	82.0 (0.25)	
**Sex, n (%)**				<0.0001
Male	848 (37.2)	1420 (50.3)	1495 (48.1)	
Female	1432 (62.8)	1403 (49.7)	1595 (52.0)	
**Education, n (%)**				<0.0001
Primary/none	407 (17.9)	861 (30.5)	1236 (40.3)	
Secondary	972 (42.6)	1180 (41.8)	1111 (36.2)	
Third/higher	901 (39.5)	781 (27.7)	720 (23.5)	
**Current employment, n (%)**				<0.0001
Employed	1131 (49.6)	1111 (39.4)	692 (22.5)	
Retired	557 (24.4)	943 (33.4)	1545 (50.3)	
Other	592 (26.0)	769 (27.2)	833 (27.1)	
**Taking anti-depressant, n (%)**				<0.0001
No	2139 (93.8)	2675 (94.8)	2806 (91.4)	
Yes	141 (6.2)	148 (5.2)	264 (8.6)	
**Smoking status, n (%)**				<0.0001
Never	1078 (47.3)	1185 (42.0)	1303 (42.4)	
Past	808 (35.4)	1009 (35.8)	1300 (42.4)	
Current	394 (17.3)	628 (22.3)	467 (15.2)	
**Alcohol problem, n (%)**				0.0105
No	1801 (87.6)	1889 (86.1)	2252 (89.9)	
Yes	255 (12.4)	306 (13.9)	253 (10.1)	
**Moderate physical activity, n (%)**				<0.0001
<5 times/week	1611 (70.7)	2026 (71.8)	2386 (77.7)	
≥5 times/week	669 (29.3)	797 (28.2)	684 (22.3)	
**Diabetes, n (%)**				<0.0001
No	2216 (97.2)	2704 (95.8)	2620 (85.3)	
Yes	64 (2.8)	117 (4.2)	450 (14.7)	

^1^ H1 = Non hypertensive, ^2^ H2 = Hypertensive without medication, ^3^ H3 = Hypertensive with medication.

**Table 2 jcm-09-03735-t002:** Comparison of socio-demographic and health characteristics of participants at Wave 3 using analysis of variance (ANOVA).

	H1 ^1^	H2 ^2^	H3 ^3^	*p*-Value
**Age, years** (Mean, SE)	63.5 (0.16)	66.7 (0.20)	70.6 (0.18)	<0.0001
*n*	*1860*	*1646*	*2742*	
**Blood pressure** (Mean, SE)				<0.0001
Systolic (mm Hg)	121.5 (0.25)	152.6 (0.44)	138.1 (0.41)	
Diastolic (mm Hg)	76.0 (0.17)	90.7 (0.29)	80.5 (0.23)	
**Sex, n (%)**				<0.0001
Male	702 (377)	787 (47.8)	1344 (49.0)	
Female	1158 (62.3)	859 (52.2)	1398 (51.0)	
**Education, n (%)**				<0.0001
Primary/none	306 (16.5)	445 (27.0)	936 (34.1)	
Secondary	732 (39.4)	682 (41.4)	1039 (37.9)	
Third/higher	821 (44.2)	519 (31.5)	767 (29.0)	
**Current employment, n (%)**				<0.0001
Employed	774 (41.6)	604 (36.8)	548 (20.0)	
Retired	700 (37.7)	688 (42.0)	1585 (57.9)	
Other	385 (20.7)	348 (21.2)	605 (22.1)	
**Taking anti-depressant, n (%)**				<0.0001
No	1694 (91.1)	1529 (92.9)	2425 (88.4)	
Yes	166 (8.9)	117 (7.1)	317 (11.6)	
**Smoking status, n (%)**				<0.0001
Never	903 (48.6)	718 (43.8)	1200 (43.8)	
Past	741 (39.8)	665 (40.6)	1229 (44.8)	
Current	216 (11.6)	257 (15.7)	313 (11.4)	
**Alcohol problem, n (%)**				0.0105
No	1420 (87.1)	1073 (86.6)	1884 (89.0)	
Yes	210 (12.9)	166 (13.4)	232 (11.0)	
**Moderate physical activity, n (%)**				<0.0001
<5 times/week	1415 (76.1)	1272 (77.3)	2224 (81.1)	
≥5 times/week	445 (23.9)	374 (22.7)	518 (18.9)	
**Diabetes, n (%)**				<0.0001
No	1795 (96.5)	1578 (95.9)	2305 (84.1)	
Yes	65 (3.5)	68 (4.1)	437 (15.9)	

^1^ H1 = Non hypertensive, ^2^ H2 = Hypertensive without medication, ^3^ H3 = Hypertensive with medication.

**Table 3 jcm-09-03735-t003:** ANCOVA results of the effects of hypertensive status on different cognitive measures controlling for age and education (partial model).

	Wave 1				Wave 3				Longitudinal			
Source	F	df	*p*-Value	η2/ηp2	F	df	*p*-Value	η2/ηp2	F	df	*p*-Value	η2/ηp2
**MMSE**												
*Overall model*	291.61	5	<0.0001	0.2	275.23	5	<0.0001	0.18	8.72	5	<0.0001	0.01
Hypertensive status ^1^	0.12	2	0.89	<0.01	6.98	2	0.0009	<0.01	0.54	2	0.5806	<0.01
Age	304.27	1	<0.0001	0.05	488.15	1	<0.0001	0.07	20.94	1	<0.0001	<0.01
Education	367.74	2	<0.0001	0.11	224.17	2	<0.0001	0.07	14.97	2	<0.0001	0.01
*Residual*		5875				6116				3434		
**MoCA**												
*Overall model*	353.75	5	<0.0001	0.23	317.74	5	<0.0001	0.24	20.48	5	<0.0001	0.03
Hypertensive status ^1^	1.19	2	0.3052	<0.01	0.6	2	0.5957	<0.01	0.07	2	0.9292	<0.01
Age	427.52	1	<0.0001	0.07	592.33	1	<0.0001	0.11	79.48	1	<0.0001	0.02
Education	412.95	2	<0.0001	0.23	258.62	2	<0.0001	0.09	0.02	2	0.9756	<0.01
*Residual*		5850				4891				3179		
**CTT-1**												
*Overall model*	437.59	5	<0.0001	0.28	297.07	5	<0.0001	0.23	14.73	5	<0.0001	0.02
Hypertensive status ^1^	1.03	2	0.3579	<0.01	0.74	2	0.4752	<0.001	0.64	2	0.526	<0.01
Age	1181.83	1	<0.0001	0.17	904.83	1	<0.0001	0.16	58.3	1	<0.0001	0.02
Education	163.5	2	<0.0001	0.05	83.92	2	<0.0001	0.03	0.83	2	<0.0001	<0.01
*Residual*		5759				4915				3135		
**CTT-2**												
*Overall model*	459.11	5	<0.0001	0.29	393.71	5	<0.0001	0.29	20.69	5	<0.0001	0.03
Hypertensive status ^1^	0.89	2	0.409	<0.01	1.98	2	0.1383	<0.01	2.12	2	0.1206	<0.01
Age	1173.72	1	<0.0001	0.17	1082.36	1	<0.0001	0.18	71.1	1	<0.0001	0.02
Education	217.45	2	<0.0001	0.07	159.46	2	<0.0001	0.06	0.23	2	<0.0001	<0.01
*Residual*		5688				4808				3070		

^1^ Hypertensive status categories: H1 = Non hypertensive, H2 = Hypertensive without medication, H3 = Hypertensive with medication.

**Table 4 jcm-09-03735-t004:** ANCOVA results of the effects of medication categories on different cognitive measures controlling for age and education (partial model).

	Wave 1				Wave 3				Longitudinal			
Source	F	df	*p*-Value	η2/ηp2	F	df	*p*-Value	η2/ηp2	F	df	*p*-Value	η2/ηp2
**MMSE**												
*Overall model*	65.02	5	<0.0001	0.22	64.29	5	<0.0001	0.17	1.2	5	0.3059	0.01
Medication ^1^	0.47	2	0.6247	<0.01	0.88	2	0.4168	<0.01	2.63	2	0.0727	<0.01
Age	89.45	1	<0.0001	0.07	114.2	1	<0.0001	0.07	0.25	1	0.6168	<0.01
Education	74.48	2	<0.0001	0.11	66.4	2	<0.0001	0.08	0.2	2	0.8213	<0.01
*Residual*		1149				1555				548		
**MoCA**												
*Overall model*	74.87	5	<0.0001	0.24	68.83	5	<0.0001	0.22	2.67	5	0.0213	0.03
Medication ^1^	0.33	2	0.7211	<0.01	0.17	2	0.8435	<0.01	0.16	2	0.8502	<0.01
Age	89.4	1	<0.0001	0.07	104.12	1	<0.0001	0.08	10.58	1	0.0012	0.02
Education	94.77	2	<0.0001	0.14	82.49	2	<0.0001	0.12	0.36	2	0.6969	<0.01
*Residual*		1145				1241				494		
**CTT-1**												
*Overall model*	91.65	5	<0.0001	0.29	83.25	5	<0.0001	0.25	3.39	5	0.0051	0.03
Medication ^1^	0	2	0.9972	<0.01	2.61	2	0.074	<0.01	0.04	2	0.9656	<0.01
Age	296.67	1	<0.0001	0.21	259.33	1	<0.0001	0.17	12.43	1	0.0005	0.02
Education	31.37	2	<0.0001	0.05	32.93	2	<0.0001	0.05	0.79	2	0.4564	<0.01
*Residual*		1124				1228				490		
**CTT-2**												
*Overall model*	97.05	5	<0.0001	0.31	88.63	5	<0.0001	0.27	1.83	5	0.1056	0.02
Medication ^1^	0.49	2	0.6114	<0.01	1.9	2	0.1506	<0.01	0.16	2	0.8551	<0.01
Age	306.25	1	<0.0001	0.22	259.11	1	<0.0001	0.18	0.25	1	0.6193	<0.01
Education	37.34	2	<0.0001	0.06	46.59	2	<0.0001	0.07	3.72	2	0.025	0.01
*Residual*		1106				1195				479		

^1^ Medication categories: BB = Beta Blockers, CB = Calcium-channel blockers and ACE = Angiotensin-converting enzyme inhibitors. These categories were derived from those who reported having hypertensive and taking medication (H3) and were on monotherapy.

**Table 5 jcm-09-03735-t005:** Multivariate regression results of the effects of medication status on different cognitive measures controlling for age and education.

	Wave 1				Wave 3				Longitudinal			
Source	β	SE	*p*-Value	ηp2	β	SE	*p*-Value	ηp2	β	SE	*p*-Value	ηp2
**MMSE**												
Combination therapy ^1^	0.04	0.1	0.046	<0.01	−0.01	0.08	0.988	<0.01	−0.03	0.11	0.236	<0.01
Age	−0.25	0.01	<0.0001	0.06	−0.28	0.01	<0.0001	0.08	−0.12	0.01	<0.0001	0.01
Education												
*- Secondary education*	0.25	0.12	<0.0001	0.06	0.22	0.11	<0.0001	0.04	−0.07	0.14	0.028	<0.01
*- Higher education*	0.35	0.11	<0.0001	0.1	0.28	0.1	<0.0001	0.07	−0.11	0.13	<0.0001	0.01
**Overall model**	F (4.2150) = 115.88, *p* < 0.0001, η2 = 0.20				F (4.2770) = 113.67, *p* < 0.0001, η2 = 0.18				F (4.1537) = 5.80, *p* = 0.0001, η2 = 0.02			
**MoCA**												
Combination therapy ^1^	0.05	0.15	0.012	<0.01	0.04	0.17	0.035	<0.01	0.01	0.15	0.668	<0.01
Age	−0.29	0.01	<0.0001	0.09	−0.31	0.01	<0.0001	0.11	−0.20	0.01	<0.0001	0.04
Education												
*-Secondary education*	0.24	0.19	<0.0001	0.05	0.23	0.22	<0.0001	0.05	−0.02	0.2	0.649	<0.01
*- Higher education*	0.38	0.19	<0.0001	0.12	0.33	0.21	<0.0001	0.1	0.01	0.2	0.757	<0.01
**Overall model**	F (4.2141) = 152.90, *p* < 0.0001, η2 = 0.39				F (4.2217) = 130.47, *p* < 0.0001, η2 = 0.24				F (4.1373) = 11.8, *p* < 0.0001, η2 = 0.04			
**CTT-1**												
Combination therapy ^1^	−0.04	1.33	0.04	<0.01	−0.03	1.52	0.158	<0.01	−0.04	1.37	0.161	<0.01
Age	0.43	0.08	<0.0001	0.19	0.43	0.1	<0.0001	0.19	0.2	0.1	<0.0001	0.04
Education												
*-Secondary education*	−0.17	1.68	<0.0001	0.03	−0.19	1.93	<0.0001	0.03	0.01	1.79	0.91	<0.01
*-Higher education*	−0.24	1.68	<0.0001	0.06	−0.22	1.93	<0.0001	0.05	0.03	1.76	0.365	<0.01
**Overall model**	F (4.2084) = 139.44, *p* < 0.0001, η2 = 0.28				F (4.2176) = 115.74, *p* < 0.0001, η2 = 0.27				F (4.1341) = 10.08, *p* < 0.0001, η2 = 0.04			
**CTT-2**												
Combination therapy ^1^	−0.04	1.99	0.042	<0.01	−0.03	1.93	0.115	<0.01	−0.04	1.54	0.09	<0.01
Age	0.45	0.12	<0.0001	0.21	0.42	0.12	<0.0001	0.19	0.15	0.11	<0.0001	0.02
Education												
*-Secondary education*	−0.17	2.51	<0.0001	0.03	−0.23	2.54	<0.0001	0.05	−0.03	2.22	0.462	<0.01
*-Higher education*	−0.26	2.46	<0.0001	0.06	−0.30	2.5	<0.0001	0.08	−0.04	2.12	0.298	<0.01
**Overall model**	F (4.2039) = 171.47, *p* < 0.0001, η2 = 0.30				F (4.2104) = 167.70, *p* < 0.0001, η2 = 0.29				F (4.1294) = 7.48, *p* < 0.0001, η2 = 0.03			

^1^ Medication status was derived from those who were hypertensive, taking medication (H3), and taking combination therapy. Comparison group: Medication status = monotherapy, education = primary education.

**Table 6 jcm-09-03735-t006:** Multivariate regression results of the effects of hypertension status on different cognitive measures.

	Partial Model			Full Model		
Source	β	SE	*p*-Value	β	SE	*p*-Value
**MMSE**						
Hypertension status						
*-H2*	−0.06	0.17	0.17	−0.04	0.18	0.36
*-H3*	−0.13	0.15	0.001	−0.06	0.15	0.168
NART	0.24	0.01	<0.0001	0.25	0.01	<0.0001
Hypertension status x NART						
*-H2*	0.05	0.01	0.049	0.02	0.01	0.676
*-H3*	0.14	0.01	0.135	0.06	0.01	0.182
**Overall model**	F (8.5182) = 112.03, *p* < 0.0001			F (19.4432) = 40.47, *p* < 0.0001		
**MoCA**						
Hypertension status						
*-H2*	−0.05	0.33	0.113	−0.05	0.35	0.228
*-H3*	−0.11	0.27	0.002	−0.05	0.27	0.167
NART	0.4	0.01	<0.0001	0.41	0.01	<0.0001
Hypertension status x NART						
*-H2*	0.06	0.06	0.049	0.04	0.01	0.285
*-H3*	0.11	0.11	0.001	0.05	0.01	0.178
**Overall model**	F (8.5185) = 283.68, *p* < 0.0001			F (19.4446) = 102.28, *p* < 0.0001		
**CTT-1**						
Hypertension status						
*-H2*	0.03	3.05	0.381	−0.01	3.1	0.97
*-H3*	0.12	2.64	0.332	0.08	2.74	0.073
NART	−0.15	0.07	<0.0001	−0.15	0.07	<0.0001
Hypertension status x NART						
*-H2*	−0.05	0.09	0.169	−0.03	0.09	0.373
*-H3*	−0.13	0.08	0.059	−0.10	0.08	0.114
**Overall model**	F (8.5142) = 142.92, *p* < 0.0001			F (19.4407) = 64.59, *p* < 0.0001		
**CTT-2**						
Hypertension status						
*-H2*	0.12	3.99	0.111	0.1	4.23	0.115
*-H3*	0.13	3.11	0.211	0.1	3.41	0.112
NART	−0.23	0.07	<0.0001	−0.22	0.08	<0.0001
Hypertension status x NART						
*-H2*	−0.11	0.13	0.658	−0.10	0.13	0.114
*-H3*	−0.12	0.1	0.112	−0.11	0.1	0.122
**Overall model**	F (8.5043) = 269.90, *p* < 0.0001			F (19.4342) = 96.41, *p* < 0.0001		

Comparison groups: hypertensive status = H1 (no hypertension). Note: H2 (hypertension without medication), H3 (hypertension with medication).

**Table 7 jcm-09-03735-t007:** Multivariate regression results of the effects of medication categories on different cognitive measures.

	Partial Model			Full Model		
Source	β	SE	*p*-Value	β	SE	*p*-Value
**MMSE**						
Medication categories						
*-CB*	0.13	0.42	0.117	0.18	0.41	0.046
*-ACE*	0.07	0.32	0.439	0.13	0.34	0.202
NART	0.32	0.01	<0.0001	0.31	0.01	<0.0001
Medication categories x NART						
*-CB*	−0.08	0.01	0.298	−0.13	0.02	0.123
*-ACE*	−0.03	0.01	0.723	−0.11	0.01	0.29
**Overall model**	F (8.1214) = 26.22, *p* < 0.0001			F (19.1023) = 10.16, *p* < 0.0001		
**MOCA**						
Medication categories						
*-CB*	0.06	0.86	0.472	0.07	0.82	0.393
*-ACE*	0.07	0.62	0.375	0.09	0.59	0.298
NART	0.46	0.02	<0.0001	0.44	0.02	<0.0001
Medication categories x NART						
*-CB*	−0.06	0.03	0.449	−0.09	0.03	0.282
*-ACE*	−0.06	0.02	0.446	−0.10	0.02	0.263
**Overall model**	F (8.1214) = 53.37, *p* < 0.0001			F (19.1020) = 21.02, *p* < 0.0001		
**CT-1**						
Medication categories						
*-CB*	−0.07	7.08	0.36	−0.02	7.68	0.839
*-ACE*	−0.05	6.34	0.559	−0.05	6.7	0.628
NART	−0.24	0.17	<0.0001	−0.20	0.18	0.005
Medication categories x NART						
*-CB*	0.01	0.22	0.848	−0.04	0.23	0.644
*-ACE*	0.01	0.19	0.964	−0.01	0.2	0.954
**Overall model**	F (8.1206) = 30.03, *p* < 0.0001			F (19.1014) = 17.12, *p* < 0.0001		
**CT-2**						
Medication categories						
*-CB*	−0.06	9.14	0.419	−0.05	9.52	0.525
*-ACE*	−0.09	7.59	0.279	−0.03	7.86	0.707
NART	−0.31	0.21	<0.0001	−0.26	0.22	<0.0001
Medication categories x NART						
*-CB*	0.03	0.29	0.679	0.03	0.3	0.685
*-ACE*	0.04	0.24	0.622	−0.01	0.24	0.934
**Overall model**	F (8.1175) = 59.15, *p* < 0.0001			F (19.995) = 22.76, *p* < 0.0001		

Comparison groups: medication categories = BB (Beta Blockers). Note: CB (Calcium Channel Blockers), ACE (Angiotensin-converting enzyme inhibitors).

**Table 8 jcm-09-03735-t008:** Multivariate regression results of the effects of medication status on different cognitive measures.

	Partial Model			Full Model		
Source	β	SE	*p*-Value	β	SE	*p*-Value
**MMSE**						
Medication status						
*-Combination therapy*	0.09	0.21	0.079	0.07	0.21	0.247
NART	0.34	0.01	<0.0001	0.31	0.01	<0.0001
Medication status x NART						
*-Combination therapy*	−0.12	0.01	0.024	−0.10	0.01	0.09
**Overall model**	F (6.2175) = 73.97, *p* < 0.0001			F (17.1811) = 10.16, *p* < 0.0001		
**MoCA**						
Medication status						
*-Combination therapy*	0.14	0.38	0.082	0.15	0.38	0.064
NART	0.52	0.1	<0.0001	0.51	0.1	<0.0001
Medication status x NART						
*-Combination therapy*	−0.14	0.01	0.212	−0.16	0.12	0.092
**Overall model**	F (6.2175) = 159.81, *p* < 0.0001			F (17.1803) = 46.88, *p* < 0.0001		
**CTT-1**						
Medication status						
*-Combination therapy*	−0.07	3.59	0.174	−0.04	3.48	0.413
NART	−0.25	0.09	<0.0001	−0.23	0.09	<0.0001
Medication status x NART						
*-Combination therapy*	0.04	0.11	0.377	0.04	0.11	0.525
**Overall model**	F (6.2143) = 92.92, *p* < 0.0001			F (17.1791) = 34.01, *p* < 0.0001		
**CTT-2**						
Medication status						
*-Combination therapy*	−0.13	4.61	0.006	−0.08	4.87	0.114
NART	−0.35	0.12	<0.0001	−0.32	0.12	<0.0001
Medication status x NART						
*-Combination therapy*	0.12	0.15	0.012	0.1	0.15	0.074
**Overall model**	F (6.2075) = 165.06, *p* < 0.0001			F (17.1746) = 47.34, *p* < 0.0001		

Comparison groups: medication status = Monotherapy.

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
