# Peer review of "Associations between Hypertension, Treatment, and Cognitive Function in the Irish Longitudinal Study on Ageing"

_jcm, 2020, doi:10.3390/jcm9113735_

Round 1

Reviewer 1 Report

The authors have appropriately addressed my critiques. 

Author Response

Comment:

The authors have appropriately addressed my critiques. 

Response:

Thank you for approving the manuscript.

Reviewer 2 Report

Journal of Clinical medicine

Manuscript ID: JCM-91370

REVISION 2

Associations between hypertension, treatment, and 2 cognitive function in the Irish Longitudinal Study on Ageing.

Authors: Sook Ling Leong, Ian H. Robertson, Brian Lawlor, and Sven Vanneste

Some general comments

Many of my concerns have been addressed in this revised version. However, suggest another revision of the method and result section to make findings clearer and transparent. An overall comment is that most of the tables could be attached as supplementary material while the main findings are nil findings. NB still the results are of large interest for all people that uses hypertension medication but they have to be outspoken more transparent and clear.

Points to revise:

1) line 116, MMSE cut-off point below 24 points indicate cognitive problems.

2) line 143, what are the independent and the dependent variables? What are possible confounding variables (to control for) Please clarify, as a suggestion through an informative figure.

3) line 151, what are the outcome measures? Please clarify, maybe in the figure suggested above.

4) about the results, As the tables are large and quite uninformative (they mainly just show non-significant effects of hypertensive status and medication) they could be removed to supplementary materials. This would simplify the presentation and increase readability.

5) Table 1 from first draft. First – please put back the table 1 from the previous manuscript, it was very informative about all necessary participant characteristics.  

6) Table 1 – draft 2.

  1. a) Hypertensive status is indeed misleading – there are three groups – i) non-hypertensive; ii) hypertensive with medication; iii) hyper tensive without medication. This has to be spoken out – other vice one gets the impression that all participants are hypertensive.
  2. b) It is not clear how the longitudinal effect is calculated? Is it a repeated measures ANCOVA? If so why is not time x group interaction reported?
  3. c) rows in the table are not synced df – impossible to interpret
  4. d) first line in the results – How can a F-value as low as 0.12 yield significance <0.0001 ? must be wrong

7) Table 2. draft 2

  1. a) What is the outcome measure of longitudinal here? Is it value at Wave 3 minus the value at Wave 1? Please clarify.
  2. b) the effect size column (eta2) is out of sync
  3. c) even in the supplementary material you do not need to have both Beta and t-values, Beta values are sufficient.

8) Table 3, draft 2

  1. a) even here the columns are out of sync.
  2. b) Here you have a variable medication – does this variable only consist of the two medicated groups? Please clarify.
  3. c) make the same clarifications of longitudinal in this table as well

9) Table 4. draft 2

  1. a) even here the columns are out of sync.
  2. b) Here a new variable pops up – medication category, is this the same as medication in table 3? Are there 2 or 3 groups? Please clarify.
  3. c) make the same clarifications of longitudinal in this table as well

10) Table 5 and 6, draft 2 (here the columns are synced)

  1. a) The variable Combination therapy, explain how these groups are formed?
  2. b) make the same clarifications of longitudinal in this table as well

11) Table 6 second table and 7, draft 2

  1. a) There are two table 6, correct numbers
  2. b) Clarify group abbreviation H2 and H3, are non-hypertensive excluded?
  3. c) explain why NART is taken as a predictor

General comment.

12) In the method section you should make all dependent, independent, and confounding variables much clearer. Moreover, write a paragraph of your analytic strategy, what assessments are made to answer what questions?

13) even in supplementary tables – in order to make those more transparent – don’t use all the output you receive from your statistical program, remove redundant information. You could also split up the results from the cognitive tests into separate tables, so each table does not exceed one page: MMSE; MOCA; and CT1 and CT2 in three separate tables.

14) write out all the results in running text in the result section and add where necessary central statistical values within brackets.

Author Response

Response to review

Manuscript ID: JCM-91370

REVISION 2

Associations between hypertension, treatment, and 2 cognitive function in the Irish Longitudinal Study on Ageing.

Authors: Sook Ling Leong, Ian H. Robertson, Brian Lawlor, and Sven Vanneste

Some general comments

Many of my concerns have been addressed in this revised version. However, suggest another revision of the method and result section to make findings clearer and transparent. An overall comment is that most of the tables could be attached as supplementary material while the main findings are nil findings. NB still the results are of large interest for all people that uses hypertension medication but they have to be outspoken more transparent and clear.

Points to revise:

Comment:

1) line 116, MMSE cut-off point below 24 points indicate cognitive problems.

Response:

Thank you for pointing out this mistake. We have changed the sentence to:

‘MMSE has a maximum score of 30 with a of cut-off of below 24 indicating cognitive problems.’

Comment:

2) line 143, what are the independent and the dependent variables? What are possible confounding variables (to control for) Please clarify, as a suggestion through an informative figure.

Response:

Thank you for the suggestion. We have included a description of the independent and dependent variables in the methods section (statistical analyses) and incorporated a figure (Figure 1) for clarification.

Comment:

3) line 151, what are the outcome measures? Please clarify, maybe in the figure suggested above.

We have included a figure (Figure 1) listing the independent, dependent, confounding and interaction variables.

Comment:

4) about the results, As the tables are large and quite uninformative (they mainly just show non-significant effects of hypertensive status and medication) they could be removed to supplementary materials. This would simplify the presentation and increase readability.

Response:

We have moved all the full models to the supplement, keeping only the partial models (controlled for age and education) in the main manuscript.

Comment:

5) Table 1 from first draft. First – please put back the table 1 from the previous manuscript, it was very informative about all necessary participant characteristics.  

Response:

We have included table 1 and 2 from the first draft in this version of the manuscript.

Comment:

6) Table 1 – draft 2.

  1. a) Hypertensive status is indeed misleading – there are three groups – i) non-hypertensive; ii) hypertensive with medication; iii) hyper tensive without medication. This has to be spoken out – other vice one gets the impression that all participants are hypertensive.
  2. b) It is not clear how the longitudinal effect is calculated? Is it a repeated measures ANCOVA? If so why is not time x group interaction reported?
  3. c) rows in the table are not synced df – impossible to interpret
  4. d) first line in the results – How can a F-value as low as 0.12 yield significance <0.0001 ? must be wrong

Response:

  1. a) A full description of the different hypertensive groups can in found in the first paragraph of the methods under statistical analysis.
  2. b) The change in cognitive assessments from W1 to W3 were used in the longitudinal in ANCOVA models. We have also included an explanation in paragraph 2 of the statistical analysis section in the methods.
  3. c) We apologize for the rows that were not synced. There was an issue when words converted to pdf.
  4. d) The rows are in sync and this should not be an issue.

Comment:

7) Table 2. draft 2

  1. a) What is the outcome measure of longitudinal here? Is it value at Wave 3 minus the value at Wave 1? Please clarify.
  2. b) the effect size column (eta2) is out of sync
  3. c) even in the supplementary material you do not need to have both Beta and t-values, Beta values are sufficient.

Response:

  1. a) The change in cognitive assessments from W1 to W3 were used in the longitudinal in ANCOVA models. We have also included an explanation in paragraph 2 of the statistical analysis section in the methods.
  2. b) We have corrected this.
  3. c) We have removed the t-value and only included the Beta values.

Comment:

8) Table 3, draft 2

  1. a) even here the columns are out of sync.
  2. b) Here you have a variable medication – does this variable only consist of the two medicated groups? Please clarify.
  3. c) make the same clarifications of longitudinal in this table as well

Response:

  1. a) We have corrected this.
  2. b) We have clarified this below the tables as well as in the first paragraph of the statistical analysis section of the methods.
  3. c) We have made the clarifications on how the dependent outcome of the longitudinal models were obtained in the statistical analysis section.

Comment:

9) Table 4. draft 2

  1. a) even here the columns are out of sync.
  2. b) Here a new variable pops up – medication category, is this the same as medication in table 3? Are there 2 or 3 groups? Please clarify.
  3. c) make the same clarifications of longitudinal in this table as well

Response:

  1. a) We have corrected this.
  2. b) We have clarified the differences in between hypertension status, medication categories and medication status in the statistical analysis section as well as under the tables.
  3. c) We have made the clarifications on how the dependent outcome of the longitudinal models were obtained in the statistical analysis section.

Comment:

10) Table 5 and 6, draft 2 (here the columns are synced)

  1. a) The variable Combination therapy, explain how these groups are formed?
  2. b) make the same clarifications of longitudinal in this table as well

Response:

  1. a) We have now added an explanation of combination therapy in the statistical analysis section (paragraph 1).
  2. b) We have made the clarifications on how the dependent outcome of the longitudinal models were obtained in the statistical analysis section.

Comment:

11) Table 6 second table and 7, draft 2

  1. a) There are two table 6, correct numbers
  2. b) Clarify group abbreviation H2 and H3, are non-hypertensive excluded?
  3. c) explain why NART is taken as a predictor

Response:

  1. a) We apologize for the mistake, we have corrected this
  2. b) Non-hypertensives were use as the comparison group in the longitudinal models. We have included legends below the tables clarifying this.
  3. c) We have included an explanation for using NART in the interaction model. This can be found in paragraph 2 under the hypertension and cognitive function paragraph.

General comment.

Comment:

12) In the method section you should make all dependent, independent, and confounding variables much clearer. Moreover, write a paragraph of your analytic strategy, what assessments are made to answer what questions?

Response:

We have re-written the statistical analyses section incorporating why each models were built. In addition, we have included a figure (Figure 1) clarifying the statistical approaches as well as the different dependent, independent and confounding variables.

Comment:

13) even in supplementary tables – in order to make those more transparent – don’t use all the output you receive from your statistical program, remove redundant information. You could also split up the results from the cognitive tests into separate tables, so each table does not exceed one page: MMSE; MOCA; and CT1 and CT2 in three separate tables.

Response:

We have removed the t values from the multivariate models (full models). We have ensured that the tables would fit into one page.

Comment:

14) write out all the results in running text in the result section and add where necessary central statistical values within brackets.

Response:

We have included important information in the running text referencing to the different tables in the manuscript and supplementary material.

This manuscript is a resubmission of an earlier submission. The following is a list of the peer review reports and author responses from that submission.

Round 1

Reviewer 1 Report

Manuscript  ID: JCM-91370

Associations between hypertension, treatment, and 2 cognitive function in the Irish Longitudinal Study on 3 Ageing. (Stage 1)

Authors: Sook Ling Leong, Ian H. Robertson, Brian Lawlor, and Sven Vanneste

Summary

This paper investigates the effects of hypertension and three hypertension treatments on cognition in a large sample (8173) of people over 50 years of age from the Irish Longitudinal Study on Ageing (TILDA). Comparisons of cognitive performance were made between 3 groups, 1) non-hypertensive (mean age 59,5), 2) non-medicated hypertensive (mean age 62,7) and 3) medicated hypertensive (mean age 68,1). A standard test battery comprising four cognitive tests, MMSE, MoCA, CTT, and NART were used. These tests are normally used to diagnose dementia or mild cognitive impairment (MCI). Results showed no effects on cognitive performance of neither hypertension nor hypertensive medication, a nil finding.

Overall comment

The present RQ is interesting and relevant, in particular with a worldwide ageing population displaying increased levels of welfare diseases such as hypertonia and diabetes type 2. Moreover, the study sample from the Irish Longitudinal Study on Ageing (TILDA) with over 8’ participants is impressive. Despite this I find the present study problematic due to three major problems 1) the huge difference in age between the three groups 2) the test battery, MMSE, MoCA, CTT, and NART, is primarily made to capture dementia and/or MCI. This implies a large risk for ceiling effects and in its prolonging type 2 errors. I realise that researchers are stuck with the TILDA test battery, so it is not so easy to solve this problem. My suggestion will be to isolate the two confounding factors, age and education, while they override the target variables, hypertension and medication and use them as covariates in ANOVA’s instead. 3) there are many confounding variables not taken into account, e.g. lifestyle variables. Moreover, most of the confounding variables were just dichotomized (yes or no and not continuous), which makes these independent variables blunt and hard to estimate effects from. These three major problems should be mentioned and exemplified in the introduction and in the discussion. Followingly, the risk for type 2 errors due this should also be mentioned both in the abstract and in the conclusions. Below, more detailed major and minor concerns are enlisted.

Major points

  1. In the introduction when education, cognitively demanding jobs are mentioned nothing is said about lifestyle factors like: smoking, eating habits, physical activity etc etc. A lot is written about successful aging (and the opposite) – check studies from e.g. the Betula project, Umeå, Sweden (Nilsson et al., 2004; Nilsson et al., 1997).

  2. Referring to the point above, if data on cognitive decline in hypertension this provides compelling arguments to control for possible confounding variables, such as lifestyle and various habits. If it is possible have these continuous the effects of each could easily be partialled out.

  3. table 1 about participants, the age differences between the three groups are huge, which will make comparisons between groups more or less unmanageable. The large effect of age will overshadow the possible effect of medication or group. Instead of doing separate ANOVA’s and regressions I suggest an ANOVA with age (and education) as covariates for each cognitive test.
  4. the non-hypertensive group is almost 10 yrs younger than the medicated sample which makes employment status rather meaningless, the hypertensive participants are of course retired to a very high extent – but what did their earlier work life look like. To be employed is a protective factor in many regards. This should at least be mentioned in the discussion

  5. table 1. Antidepressants is just a yes /no question, for how long, and the dose is relevant when it comes to cognition, same goes for smoking status, alcohol problem, and diabetes (medication: tablets or injections, type 1 or type 2 diabetes). The same remarks goes for table 2 as well.

  6. Figure 1-3. Age should be taken as a covariate and Bonferroni corrections should be made. When you take age and education into account all the effects of group (H1-3) disappears I guess, then your data shows that age and education affect cognition, and this is known since long. The same reasoning goes for figure 2 and 3 as well. To sum it all up – what new knowledge does these data provide about hypertension, medication and cognition that was the main goal of the present study as I read it. The graphs 1-3 could be replaced with a table instead – while there are no effects (or interactions) the figures do not add any new information.

  7. Test battery, The MMSE and MoCA tests are in particular made to detect dementia (or MCI), these 10 min test may not be sensitive enough to capture changes, risk for ceiling effects as well du to small variability cut of is 26 of 30 to receive a diagnosis. CTT can also be made on younger populations but it has a significant contribution of age and educational level in both parts of the CTT (e.g. Messinis et al, 2011) and this was not the main purpose of the present study, but rather the effects of medication. Also, the NART test is mostly used in clinical settings for estimating premorbid intelligence (low IQ). So taken together, most of the test battery is adapted for clinical groups in the domain of cognitive functioning and the question here is whether the hypertensive group is a good representative of MCI or cognitively defined clinical groups, a more sensitive test battery with a broader performance range would have been preferable in this case. This cannot be changed in retrospect but the possibility of type 2 error due to this should be mentioned both in the abstract and the discussion.

  8. The fact that larger effects were found in CCT 1, 2 is probably due to diminished risk of ceiling effects, this comprise a larger span between high and low performers.

  9. Figure 4-6, however, that measures effects within the medicated group provides new information. It looks like Beta- blockers have the least impact on cognition s compared with CB and ACE medication (post hoc test with Bonferroni corrections should be made here). But again, does the significance survive an ANOVA with age and education as covariates? I urge that you add these assessments in the result section and consider if not a table better could summarize the findings from figures 4-6.

  10. The large effect of education should be problematized in the discussion whether this is an effect of selection (only those with high cognitive performance chose and fulfil education) or a cause, that education per se has positive effects on cognition.

Minor points

1. Line 28, dementia above 80 million - in Europe?

2. Line 30, MCI and AD should be out spelled the first time

3) Line 49, a review of 358 is a meta-analysis I guess

4) Discussion first paragraph, participants who were on BB, CB, or ACS medication (add medication)

Kind regards
Reviewer 1

Nilsson, L.-G., Adolfsson, R., Bäckman, L., De Frias, C. M., Molander, B., & Nyberg, L. (2004). Betula: A Prospective Cohort Study on Memory, Health and Aging. Aging, Neuropsychology, and Cognition, 11(2-3), 134-148. doi:10.1080/13825580490511026

Nilsson, L.-G., Bäckman, L., Erngrund, K., Nyberg, L., Adolfsson, R., Bucht, G., . . . Winblad, B. (1997). The Betula prospective cohort study: Memory, health, and aging. Aging, Neuropsychology, and Cognition, 4(1), 1-32. doi:10.1080/13825589708256633

Reviewer 2 Report

This study uses a large dataset to examine the associations between hypertension, antihypertensive medications, and cognition. As previous reports have been conflicting, further research on this topic is warranted. There are issues related to the statistical analyses (e.g. use of univariate analyses, separate groups for individuals with hypertension depending on medication status) and presentation of the results that need to be addressed to improve the interpretation of the findings.

  1. The third paragraph of the introduction provides a synopsis of the prior literature. While the first sentence states that there are “conflicting results regarding hypertension and the development of cognitive decline”, several of the studies described report on blood pressure as a continuous measure. Please be sure to clarify and separately highlight results for blood pressure as a continuous variable vs hypertension.
  2. Paragraph 5 of the introduction abruptly changes the discourse from the cognitive effects of hypertension to cognitive reserve. Please provide a more thorough synopsis of the literature about cognitive reserve specific to its association with hypertension.  Also, since there are many different definitions of cognitive reserve, it would be helpful to introduce the reader to specific variables related to cognitive reserve that will be examined in the manuscript.
  3. Line 83 – please explain the rationale for including both the MMSE and MoCA. Also please clarify if the two measures were collected at the same timepoint. If the measures were collected at the same timepoint, please explain the procedures used to address overlapping variables on both measures (i.e. orientation questions) and precautions taken to prevent interference from the two word lists (one on MMSE and one on MoCA) on delayed recall.
  4. In Table 2, please also add information on cognitive test performance and antihypertensive medication categories.
  5. Rather than separately examining hypertension with and without medication, the main question of the impact of hypertension on cognition would be better addressed by collapsing these two groups. Please re-run analyses with a combined hypertension group. Supplemental analyses can be used to examine the impact of antihypertensive medication on cognition.
  6. Multivariate regression was only conducted when univariate models were significant. This approach is problematic since univariate associations may be masked by confounders (such as the very powerful effects of age and education on cognition). Please omit the univariate analyses and conduct multivariate regression for all outcomes with adjustment for multiple comparisons.
  7. Line 117 – Why were interactions with monotherapy only examined for the MMSE? The MMSE lacks sensitivity so the interaction should also be examined with color trails.
  8. Line 118 – Why was heteroskedasticity assumed? What procedure was used to ensure that correction was appropriate.
  9. It is challenging to integrate all the findings presented in the figures. It would be helpful to display the results in table form with figures to showcase significant findings as based on the multivariate analyses.